# The Greatwall–Endosulfine Switch Accelerates Autophagic Flux during the Cell Divisions Leading to G1 Arrest and Entry into Quiescence in Fission Yeast

**DOI:** 10.3390/ijms24010148

**Published:** 2022-12-21

**Authors:** Alicia Vázquez-Bolado, Rafael López-San Segundo, Natalia García-Blanco, Ana Elisa Rozalén, Daniel González-Álvarez, M. Belén Suárez, Livia Pérez-Hidalgo, Sergio Moreno

**Affiliations:** 1Instituto de Biología Funcional y Genómica (IBFG), CSIC, Salamanca University, Zacarías González 2, 37007 Salamanca, Spain; 2Department of Microbiology and Genetics, Salamanca University, 37007 Salamanca, Spain

**Keywords:** autophagy, quiescence, Greatwall, Endosulfine, PP2A, fission yeast

## Abstract

Entry into quiescence in the fission yeast *Schizosaccharomyces pombe* is induced by nitrogen starvation. In the absence of nitrogen, proliferating fission yeast cells divide twice without cell growth and undergo cell cycle arrest in G1 before becoming G0 quiescent cells. Under these conditions, autophagy is induced to produce enough nitrogen for the two successive cell divisions that take place before the G1 arrest. In parallel to the induction of autophagy, the Greatwall–Endosulfine switch is activated upon nitrogen starvation to down-regulate protein phosphatase PP2A/B55 activity, which is essential for cell cycle arrest in G1 and implementation of the quiescent program. Here we show that, although inactivation of PP2A/B55 by the Greatwall–Endosulfine switch is not required to promote autophagy initiation, it increases autophagic flux at least in part by upregulating the expression of a number of autophagy-related genes.

## 1. Introduction

Autophagy is a highly conserved catabolic process, from yeast to humans, that helps maintain cellular homeostasis by recycling cellular components in nutrient- or energy-deficient conditions [1,2,3]. During autophagy, cytoplasmic components and organelles (ribosomes, mitochondria, etc.) are incorporated into double-membrane vesicles or autophagosomes [3,4] and delivered to the degradative compartment of the cell, the vacuole in yeast, or the lysosome in animal cells, where acid hydrolases degrade the autophagosome contents before being recycled to the cytoplasm and used in biosynthetic pathways.

Autophagosome formation is a multi-step process mediated by a conserved group of proteins encoded by autophagy-related genes (ATG), first discovered in the budding yeast *Saccharomyces cerevisiae* [1,5]. The initial step of this pathway is regulated by the Atg1 complex in yeast [6,7] or the Ulk1 complex in mammalian cells [8,9,10]. Autophagy is normally activated by nutritional stress signals. The Target of Rapamycin (TOR) kinase, which plays a key role in promoting cell growth, inhibits autophagy. TOR is a serine/threonine protein kinase that is part of two complexes, TOR complex 1 (TORC1) and TOR complex 2 (TORC2), with conserved functions throughout evolution. Phosphorylation of Atg13, part of the Atg1 complex, by TORC1 prevents the activation of the Atg1 complex in budding and fission yeasts [11,12] and animal cells [13,14], thus blocking autophagy under nutrient-rich conditions.

In fission and budding yeasts, TORC1 also inhibits the Greatwall–Endosulfine switch that operates as an inhibitor of the protein phosphatase PP2A/B55. TORC1 phosphorylates and activates the S6 kinases, Sck1, Sck2, and Psk1, in fission yeast, which phosphorylate and inhibit Greatwall (Ppk18, Cek1, and Ppk31 in fission yeast, Rim15 in budding yeast). Thus, in nitrogen-rich media, when TORC1 activity is high, the Greatwall–Endosulfine (Ppk18, Cek1, and Ppk31-Igo1) switch is OFF and, as a result, the PP2A/B55 (PP2A/Pab1) phosphatase has high activity. Conversely, in nitrogen-poor media, TORC1 activity decreases, allowing the activation of Greatwall (Ppk18, Cek1, and Ppk31), which phosphorylates Endosulfine (Igo1) to become a potent inhibitor of the protein phosphatase PP2A/B55. Low levels of PP2A/B55 phosphatase activity allow cells to accelerate entry into mitosis and to divide twice without cell growth before arresting in G1 and entering quiescence [15,16] or initiating the sexual differentiation response [17,18].

Here we show that in fission yeast cell cycle exit, autophagy, G1 arrest, and entry into quiescence are coordinated by the activity of the protein phosphatase PP2A/B55 through the Greatwall–Endosulfine switch, which is under the control of TORC1. During nitrogen starvation, the inactivation of TORC1 and S6 kinases leads to the activation of the Greatwall–Endosulfine switch, which turns OFF PP2A/B55 activity. Inactivation of PP2A/B55 is required to increase the autophagic flux and to promote the acceleration of the G2/M transition during the last two cell divisions without cell growth, leading to a decrease in cell size and G1 arrest.

## 2. Results

### 2.1. The Greatwall–Endosulfine-PP2A/B55 Pathway Regulates Proper G1 Arrest under Nitrogen Starvation

In fission yeast, the master regulator of cell growth TORC1 controls the Greatwall–Endosulfine–PP2A/B55 (Ppk18,Cek1-Igo1-PP2A/Pab1) pathway that functions as a molecular switch to coordinate cell growth with cell size, quiescence, and cell differentiation [15,16,17,18]. In nitrogen-rich media, TORC1 activity is elevated and phosphorylates the S6 kinase orthologues, Sck1, Sck2, and Psk1, which in turn phosphorylate and inhibit Greatwall (Ppk18 and Cek1), a protein kinase that phosphorylates Endosulfine (Igo1), a small protein that in its phosphorylated form acts as a potent and specific inhibitor of the PP2A/B55 (PP2A/Pab1) complex ([15]; Figure 1A). Thus, in nitrogen-rich medium, when TORC1 (and S6) kinase activity is high, the Greatwall–Endosulfine (Ppk18, Cek1, and Ppk31-Igo1) switch is OFF and PP2A/B55 (Ppa1-Pab1 and Ppa2-Pab1) activity is high (Figure 1A, top). In contrast, in nitrogen-poor media, when TORC1 (and S6) activity decreases, Greatwall (Ppk18 and Cek1) is activated and phosphorylates Endosulfine (Igo1) on Serine 64. Phosphorylated Ser64-Igo1 binds to and inhibits the PP2A/B55 phosphatase complex (Figure 1A, bottom). As a consequence, PP2A/B55 phosphatase activity in nitrogen-poor media is reduced, cells divide at a smaller size and undergo proper G1 arrest and entry into quiescence after two successive cell divisions ([15,16]; Figure 1B). Thus, mutants lacking Endosulfine (*igo1*∆) or Greatwall (*ppk18*∆ *cek1*∆) were unable to accelerate entry into mitosis, did not reduce cell size at division, and did not undergo cell cycle arrest in G1 in response to nitrogen starvation (Figure 1C). The *igo1*∆ and *ppk18*∆ *cek1*∆ mutants divided only once and arrested the cell cycle in G2 (Figure 1C; [15,16]). In contrast, moderate overexpression of *ppk18*^+^ generated cells with a reduced cell size that were delayed in G1 ([15]; Figure 1C, *nmt41:ppk18* ON). These small cells divided only once and arrested the cell cycle in G1 after nitrogen starvation (Figure 1C, *nmt41:ppk18* ON).

The *igo1*∆ *ppa2*∆ or *igo1*∆ *nmt41:pab1* double mutants, which lack the main catalytic subunit of PP2A (Ppa2) or express reduced levels of the B55 regulatory subunit (Pab1) in the presence of thiamine (promoter OFF), respectively, rescued the phenotype of the *igo1*∆ mutant (Figure 1D). Upon nitrogen starvation, the double mutants arrested in G1 displayed phenotypes similar to those of wild type, *ppa2*∆ or *nmt41:pab1 OFF* single mutants ([16]; Figure 1D), suggesting that high levels of PP2A/Pab1 activity are responsible for the inability of the *igo1*∆ mutant to arrest in G1 in response to nitrogen starvation. Interestingly, this phenotype shows similarities to nitrogen-starved autophagy mutants in fission yeast ([19]; Appendix A). In conclusion, Greatwall (*ppk18*∆ *cek1*∆), Endosulfine (*igo1*∆), and autophagy mutants (*atg2*∆, *atg5*∆, *atg13*∆, and *atg1802*∆), which block autophagy at different stages, arrest the cell cycle mainly in G2 after a single cell division following nitrogen starvation (Figure 1B,C; Appendix A).

### 2.2. TORC1 and Greatwall Protein Kinase Activity Oscillates during Nitrogen Starvation

Nitrogen-starved fission yeast cells rapidly inactivate TORC1. Therefore, Greatwall is activated, promoting the phosphorylation of Igo1 at Ser64, which can be monitored using specific antibodies that recognize the conserved phospho-Ser64 epitope. Figure 2 shows that the levels of phosphorylated Igo1 at Ser64 increased in wild type cells, reaching a maximum in 30 min before decreasing after 4 h of nitrogen starvation. This oscillation in Greatwall activity mirrored TORC1 activity (Figure 2, Rps6-P signal). This result clearly indicates that both TORC1 and Greatwall activity oscillate out of phase during the two cell divisions that occur after nitrogen starvation. The transient reactivation of TORC1 after 4 h of nitrogen starvation could be explained by the generation of recycled nitrogen, mainly amino acids, in the vacuole by autophagy (see Section 2.3, Figure 3). Reactivation of TORC1 inactivated Greatwall (Figure 2, time 4 h). TORC1 reactivation also occurred in mutants lacking Endosulfine (*igo1*∆), but in this case, with a delay of about 4 h (Figure 2, time 8 h). A possible interpretation of this result could be that the onset of autophagy is delayed in the *igo1*∆ mutant compared to the wild type (see Section 2.3, Figure 3). This experiment clearly indicates that in fission yeast, TORC1 and Greatwall protein kinase activities oscillate out of phase during the two cell divisions that precede G1 arrest after nitrogen starvation.

### 2.3. The Greatwall (Ppk18, Cek1 and Ppk31)–Endosulfine (Igo1) Switch Accelerates Autophagic Flux

To test whether the Ppk18, Cek1, and Ppk31-Igo1 switch play a role in the regulation of autophagy, we decided to assess autophagic flux in different mutants of this pathway using the processing of N-terminally-tagged Atg8 with CFP as a readout of autophagy. CFP is resistant to protein degradation in the vacuole, and thus, when autophagy is induced, a free CFP band can be detected with Western blot [20,21,22].

Autophagy was induced by transferring the fission yeast cells from minimal medium (MM) to minimal medium without nitrogen (MM-N) [21,22]. Samples were collected at different times for Western blot analysis. CFP:Atg8 was processed to free CFP in CFP:Atg8 wild type cells after 2–4 h in MM-N (Figure 3, wt). In cells lacking Endosulfine (*igo1*∆), the appearance of the free CFP band was delayed and less intense compared to the wild type (Figure 3, wt vs. *igo1*∆). The intensity of the CFP band in the *igo1*∆ mutant increased with time but did not reach the levels of wild type cells, suggesting that Igo1 is required to promote a normal autophagic flux under nitrogen starvation conditions. As a negative control of autophagy, we used the *atg13*∆ mutant, in which the formation of the Atg1 initiation complex is blocked (Figure 3). The delay in autophagy flux in the *igo1*∆ mutant was also observed in cells expressing Pgk1:GFP (Appendix A), which has been used as an alternative readout of autophagy in fission yeast [23].

**Figure 3 ijms-24-00148-f003:**
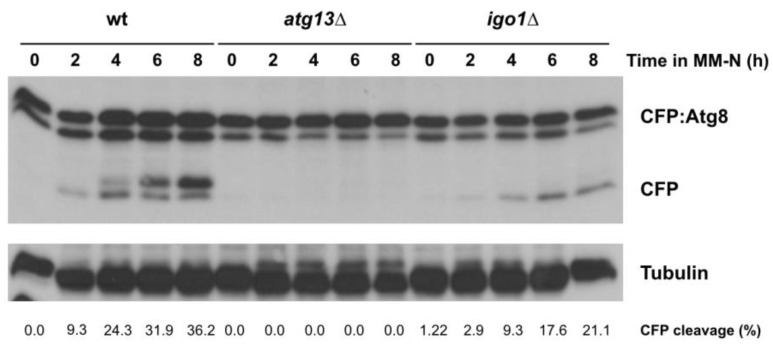
Endosulfine (Igo1) accelerates autophagic flux. Wild type, *atg13*∆, and *igo1*∆ cells expressing CFP:Atg8 were grown in minimal medium at 25 °C until mid-exponential phase and then transferred to nitrogen-free minimal medium (MM-N) for 8 h at 25 °C. Samples were taken at 0, 2, 4, 6, and 8 h in MM-N. Levels of CFP:Atg8 and free CFP were determined using anti-GFP antibodies (Clontech). Tubulin was used as a loading control. The percentage of CFP cleavage is indicated under the gels.

To establish whether Greatwall is also involved in the regulation of autophagy, we assessed the CFP:Atg8 processing in mutants of the fission yeast Greatwall orthologues Ppk18, Cek1, and Ppk31. Ppk18 is the main kinase responsible for Igo1 Ser64 phosphorylation and activation, while Cek1 plays a secondary and redundant role [15], and Ppk31 appears to play no role [16]. When autophagy is induced in these mutants, we could observe that only *ppk18Δ* showed a delay in the appearance of free CFP, a similar but less severe phenotype than the *igo1Δ* mutant (Figure 4A). In the *ppk18Δ cek1Δ* double mutant that completely lacks Greatwall activity [15], we observed a severe reduction in CFP:Atg8 processing comparable to that of *igo1*∆ (Figure 4B). The *cek1Δ ppk31*∆ double mutant also showed a slight delay in the generation of free CFP (Figure 4B). Together, these results suggest that the Greatwall kinases Ppk18 and, to a lesser extent, Cek1 and Ppk31 play a role in nitrogen starvation-induced autophagy.

To determine whether Greatwall activation by itself is capable of inducing autophagy, we monitored autophagy in cells overexpressing *ppk18^+^*. Moderate overexpression of *ppk18^+^* from the *nmt41* promoter generates small cells that show an extended G1 phase ([15]; Figure 1C). Strong overexpression of *ppk18^+^* from the *nmt1* promoter also generates small cells that fail to divide and show a phenotype similar to quiescent cells (Pérez–Hidalgo, unpublished results). Cells overexpressing *ppk18^+^* for up to 46 h in MM showed limited signs of CFP:Atg8 processing (Figure 5, left panel), indicating that strong overexpression of *ppk18^+^* in nitrogen-rich medium is not sufficient to induce autophagy. However, these cells overexpressing *ppk18^+^* for 22 h showed a slight acceleration of the autophagic flux when deprived of nitrogen (Figure 5, middle and right panels), indicating that although Greatwall activation is unable to induce autophagy in nitrogen-rich medium, it can promote an increase in autophagy flux once autophagy is induced by nitrogen starvation. Similar results were obtained when Pgk1:GFP was used as an autophagy readout (Appendix A).

Interestingly, *ppk18^+^* overexpression upregulates the expression of a number of autophagy-related genes, including *cpy1^+^*, *atg2402^+^, atg43^+^, atg1^+^*, *atg13^+^, atg1801^+^*, *atg4^+^*, *atg20^+^* and *atg7^+^* (Figure 6A and Appendix A), indicating that the Greatwall–Endosulfine-PP2A/B55 pathway may be upregulating the expression of a subset of genes encoding proteins involved in autophagy. Consistent with this idea, Greatwall (*ppk18*∆ *cek1*∆) and Endosulfine (*igo1*∆) mutants showed a reduction in the expression of autophagy response genes compared to the wild type after 1 h or 4 h of nitrogen starvation (Figure 6B and Appendix A). These results suggest that increased expression of autophagy-related genes through Greatwall activation may precondition the autophagy response to nitrogen starvation.

### 2.4. S6 Kinases Are Negative Regulators of Autophagy

Ppk18 activity is negatively regulated by TORC1 and S6K [15]. In mammalian cells, TORC1 phosphorylates and activates S6 Kinase [24]. In *S. pombe,* three proteins show homology to S6 kinases: Sck1, Sck2, and Psk1 [25]. To test whether S6 kinases regulate autophagy, we examined the processing of CFP:Atg8 in S6 kinase deletion mutants (Figure 7). The *sck1*∆, *sck2Δ,* and *psk1Δ* strains showed more free CFP compared to the wild type strain (Figure 7). This result suggests that the fission yeast S6 kinases negatively modulate autophagic flux, probably by negatively regulating Ppk18 and Cek1 kinase activity.

### 2.5. PP2A/Pab1 Negatively Regulates Nitrogen Starvation-Induced Autophagy

Finally, we tested the possible role of PP2A/B55 in autophagy as the element acting downstream in the pathway. The PP2A/Pab1 phosphatase complex consists of three subunits: the structural subunit Paa1, the catalytic subunits Ppa1 or Ppa2, and the regulatory subunit Pab1. One possible explanation for slow autophagic flux in Greatwall (*ppk18*∆ *cek1*∆) and Endosulfine (*igo1*∆) mutants could be that PP2A/Pab1 activity is not down-regulated during nitrogen starvation in these mutants. High levels of PP2A/B55 activity in nitrogen-starved cells could inhibit autophagy. To test this hypothesis, we reduced the levels of PP2A/Pab1 activity either by repressing *pab1^+^* gene expression, using cells expressing the *pab1^+^* gene from the *nmt41* promoter, or by deleting the *pab1^+^* gene. In both cases, autophagy flux increased slightly (Figure 8A), indicating that PP2A/Pab1 activity inhibits autophagy.

To confirm this result, we deleted the *ppa2^+^* gene, which encodes the main catalytic subunit of PP2A that contributes to 80% of the phosphatase activity [26]. Reduction of PP2A activity in *ppa2*∆ cells also increased autophagy flux (Figure 8B). Furthermore, deletion of *ppa2^+^* rescued the *igo1*∆ phenotype, suggesting that PP2A/Pab1 is acting downstream of Igo1 and that high levels of PP2A/Pab1 are indeed responsible for the inhibition of autophagy in the *igo1*∆ mutant (Figure 8B).

Taken together, these results point to a model in which PP2A/Pab1 protein phosphatase is acting as an inhibitor of nitrogen starvation-induced autophagy.

## 3. Discussion

Induction of autophagy in nitrogen-starved fission yeast cells is required for completion of the last two cell divisions that take place before G1 arrest and entry into quiescence in heterothallic (*h^−^* or *h^+^)* strains [19] and for sexual differentiation followed by mating, meiosis, and sporulation in homothallic (*h*^90^) strains [27]. Here we show that, following nitrogen starvation, TORC1 activity initially decreases before increasing again when nutrients (mainly amino acids) are generated by recycling proteins and organelles by autophagy. This oscillation of TORC1 activity provides enough biosynthetic activity to complete the second cell division prior to cell cycle exit and arrest in G1. In budding yeast and mammalian cells, it has also been shown that recycled nutrients generated by autophagy reactivate TORC1 [28,29].

Our data also indicate that the Greatwall–Endosulfine switch, which is activated in response to nitrogen starvation, increases autophagic flux by down-regulation of PP2A/B55 protein phosphatase activity. Greatwall and Endosulfine mutants show a decreased autophagy rate that is suppressed by reduced PP2A/B55 activity, suggesting that high phosphatase activity inhibits autophagy. Consistent with this idea, mutants of the catalytic and regulatory subunits of the PP2A/B55 protein phosphatase complex (PP2A/Pab1), *ppa2*∆ and *pab1*∆, showed increased autophagic flux compared to the wild type strain. Similarly, in budding yeast, deletion of the phosphatase regulatory subunit, *cdc55*∆, showed increased GFP:Atg8 processing when autophagy was induced during meiosis or by removing nitrogen from the medium [30].

Activation of the Greatwall–Endosulfine switch in nitrogen-rich medium by overexpression of *ppk18^+^*, which reduces PP2A/B55 activity, is not sufficient to induce autophagy. Autophagy initiation in these cells remains dependent on nitrogen starvation because active TORC1, as in budding yeast and animal cells, suppresses autophagy initiation through phosphorylation of Atg13 [12,31]. Phosphorylation of Atg13 reduces its interaction with Atg1 and Atg17-Atg31-Atg29 preventing the formation of the autophagy initiation complex in fission yeast [3,11,32,33,34]. TORC1 also inhibits Greatwall by phosphorylating Sck2 kinase, which in turn phosphorylates and inhibits Greatwall (Ppk18 and Cek1). In nitrogen-poor medium, inactive TORC1 allows dephosphorylation of Atg13 and its interaction with Atg1 and Atg17, resulting in the assembly of autophagosome initiation complex, thereby triggering autophosphorylation and autoactivation of Atg1 and the initiation of autophagy. In a parallel pathway, the inactivation of TORC1 and Sck2 leads to the activation of Greatwall, which in turn promotes the phosphorylation of Endosulfine at serine 64 and inhibition of PP2A/B55. We propose that these two pathways act in parallel, one promoting the initiation of autophagy and the other accelerating the autophagic flux.

In *S. cerevisiae,* Greatwall (Rim15) has been shown to phosphorylate and inactivate Rph1, a conserved transcriptional repressor of autophagy-related genes [35]. Phosphorylation of Rph1 following nitrogen starvation inhibits its activity, whereas overexpression of Rph1 inhibits autophagy induction. Several *atg* genes show an increase in expression level after nitrogen starvation in both fission yeast [36,37] and budding yeast [38], suggesting that transcriptional activation of certain *atg* genes is necessary to support optimal autophagy activity. In this work, we show that overexpression of Greatwall upregulates the expression of *cpy1^+^*, *atg2402^+^, atg43^+^, atg1^+^*, *atg13^+^, atg1801^+^*, *atg4^+^*, *atg20^+^, and atg7^+^*, suggesting that the Greatwall–Endosulfine–PP2A/B55 pathway may increase autophagic flux, at least in part, at the transcriptional level. Similar results have recently been reported indicating that the Sty1 MAPK stress pathway also plays a positive role in regulating autophagy, mainly at the level of transcription [23].

Rim15 also promotes autophagy during meiosis and sporulation, although the molecular mechanism has not been clarified [30]. In *S. cerevisiae*, Endosulfine proteins (Igo1 and Igo2) regulate autophagy during meiosis and sporulation. The double mutant *igo1*∆ *igo2*∆ showed reduced GFP:Atg8 processing when meiosis was induced [30]. Furthermore, Rim15 is required for autophagy induced by the simultaneous inhibition of PKA and Sch9 [20].

In addition to studying the link between autophagy and the Greatwall–Endosulfine–PP2A/B55 (Ppk18,Cek1-Igo1-PP2A/Pab1) pathway, we tested the effect on autophagy of different elements acting upstream and downstream of the pathway. In the presence of nitrogen, the S6 kinase orthologues Sck1, Sck2, and Psk1 are phosphorylated and activated by TORC1, similar to what occurs in mammalian cells. Although in mammalian cells, a relationship between S6K1 and Mastl, a Greatwall orthologue, has not been described, in *S. pombe,* Sck2 negatively regulates Ppk18 [15]. In mammalian cells, mTORC1 inhibits autophagy by phosphorylating the Ulk1/Ulk2 initiation complex, orthologue of Atg1 in yeast. An increase in autophagy has been described with the use of S6K1 inhibitors [39]. On the other hand, it appears that S6K1 might be required for autophagosome maturation when autophagy is induced by serum starvation. A lower level of S6K1 results in an altered autophagic flux in which autophagosomes are accumulated and there is a reduction in the number of lysosomes [40]. Our results point to the S6 kinase orthologues as inhibitors of the autophagic flux, as cells carrying a deletion of *sck2*^+^ or *psk1*^+^, and, to a lesser extent, *sck1*^+^ showed more CFP:Atg8 processing.

## 4. Materials and Methods

### 4.1. Fission Yeast Strains and Methods

The fission yeast strains used in this study are listed in Appendix A. Fission yeast cells were grown and manipulated genetically according to standard protocols [41]. Genetic crosses were performed on malt extract agar plates. Cells were typically grown overnight in yeast extract supplemented with adenine, leucine, histidine, lysine, and uracil (YES) and then transferred to Edinburgh minimal medium containing 93.5 mM ammonium chloride (MM). Nitrogen starvation experiments were performed by transferring cells to minimal medium without nitrogen (MM-N). Cells were grown to mid-exponential phase, centrifuged, and washed three times in MM-N. Experiments were performed at 25 °C.

For the overexpression of *ppk18+* from the *nmt1*^+^ promoter, cells were grown to mid-exponential phase in MM containing 5 µg/mL thiamine, harvested, washed twice with MM, and grown in fresh MM without thiamine at 25 °C for the times indicated in the figures. The maximum level of *ppk18+* overexpression was reached after 12–14 h of culture in MM.

### 4.2. Strain Construction

A PCR-based strategy was used to insert the GFP tag to express Pgk1 fused to GFP at the *pgk1^+^* genomic locus. Oligonucleotides with 80 bases of homology to the sequence upstream and downstream of the stop codon of the target gene were used to amplify the tag and resistance cassette sequence from plasmid pFA6a-GFP-kanMX6 and pFA6a-GFP-hphMX6. PCR products obtained using High Fidelity DNA polymerase (Roche, Basel, Switzerland) were used to transform a wild type strain *h^−^ 972* (S2726).

### 4.3. Flow Cytometry

Samples of 10^7^ cells were fixed in 70% (*v*/*v*) ethanol and then washed with a solution of 50 mM sodium citrate and resuspended in 0.5 mL of 50 mM of sodium citrate containing 0.1 mg/mL RNase A and incubated overnight at 37 °C. Subsequently, 0.5 mL of 50 mM of sodium citrate containing 4 μg/mL of propidium iodide was added. Cell suspensions were sonicated before analysis in a BD FACSCalibur Flow Cytometer (BD Biosciences, Franklin Lakes, NJ, USA). Analysis was performed using BD Cell Quest ProTM 6.0.3 (BD Biosciences, Franklin Lakes, NJ, USA).

### 4.4. Microscopy

Images were acquired with a Nikon Eclipse 90i microscope coupled to an Orca ER camera and equipped with MetaMorph software (Molecular Devices, San José, CA, USA). Images were processed and assembled with ImageJ software v1.53k.

### 4.5. Protein Extracts & WB

Protein extracts were prepared using trichloroacetic acid (TCA) extraction [42]. For Western blot, between 20–50 µg of total protein extract were run on 8–15% SDS-PAGE, transferred to PVDF Immobilon P membranes (Millipore, Burlington, MA, USA), and probed with rabbit anti-Igo1 (1:200), rabbit anti-P-Ser64-Igo1 (1:1000), rabbit anti-phospho-(Ser/Thr) Akt Substrate for the phosphorylation of Rps6 (1:1500) (Cell Signaling Technology, Danvers, MA, USA), mouse anti-GFP (1:3000) (Clontech, Mountain View, CA, USA, Figure 3, Figure 4, Figure 7 and Figure 8), rat anti-GFP (1:5000) (Chromotek, Planegg, Germany, Figure 5, Appendix A and Appendix A) and mouse anti-tubulin (1:10,000) (a gift from Dr. Keith Gull, Sir William Dunn School of Pathology, University of Oxford, UK) primary antibodies and, as secondary antibodies, NA 931 (anti-mouse IgG; Amersham, Buckinghamshire, UK), NA 934 (anti-rabbit; Amersham, Buckinghamshire, UK), and PO450 (anti-rat; Dako, Glostrup, Denmark) conjugated to horseradish peroxidase. Immunoblots were developed using Amersham ECL Western Blotting Detection Reagent (GE Healthcare, Chicago, IL, USA) or Clarity Western ECL substrate (Bio-Rad, Hercules, CA, USA). Protein quantification in the Western blots was performed using ImageJ (National Institutes of Health, Bethesda, MD, USA) gel analysis. The percentage of CFP (or GFP) cleavage was calculated by dividing the amount of free CFP (or GFP) by the total amount of CFP plus CFP:Atg8 (or GFP plus Pgk1:GFP).

### 4.6. RNA Extraction, RNA Purification, Library Preparation, and RNAseq

Wild type (S2726), *igo1*∆ (S2727), and *ppk18*∆ *cek1*∆ (S2883) cells were grown to mid-exponential phase, centrifuged and washed three times in MM-N, and cultured in MM-N at 25 °C. Then, 2 × 10^8^ cells were harvested at times 0, 1, and 4 h, washed with cold DEPC-H_2_0, and snap-frozen. RNA extraction was carried out by disrupting the cells with glass beads using RNAeasy Mini kit (Qiagen, Hilden, Germany) following the manufacturer’s instructions. RNA quality was evaluated on the Bioanalyzer 2100 (Agilent, Santa Clara, CA, USA). Library preparation, using the Illumina Ribo Zero and TruSeq Stranded, and subsequent NGS sequencing were performed using Macrogen, Seoul, Republic of Korea.

Wild type (S2726) and cells overexpressing *ppk18^+^* from the *nmt1*^+^ promoter (S3013) were grown in MM without thiamine for 20 h at 25 °C. Then, 2 × 10^8^ cells were harvested, washed with cold DEPC-H_2_O and snap-frozen. RNA extraction was carried out by disrupting the cells with glass beads in TRIzol^®^ Reagent (Invitrogen, Waltham, MA, USA) following the manufacturer’s instructions. Ribosomal RNAs were then depleted from 1.5 µg of total RNA samples using the NEBNext RNA Depletion Core kit (New England Biolabs, Ipswich, MA, USA) and *S. pombe*-specific ribosomal RNA depletion antisense probes. RNA quality and depletion efficiency were evaluated on the Bioanalyzer 2100 (Agilent). Library preparation, using the Illumina TruSeq Stranded Total RNA, and subsequent NGS sequencing were performed using Macrogen.

Sequencing quality was checked with FastQC (v 0.11.8, Babraham Bioinformatics). If necessary, adaptors were trimmed using Trimmomatic (v 0.38) [43]. Alignment was performed with HISAT2 v 2.1.0 (CCB at Johns Hopkins University) [44] using *S. pombe* reference genome from Pombase (downloaded on 30 November 2018). Samtools (v 1.9) and deepTools (v 3.3.0) were used to obtain bigWig files to visualize in IGV (v 2.4.16) and JBrowse (v 1.15.4) browsers. Read counts were obtained with featureCounts (Subread package v 1.6.3, Walter and Eliza Hall Bioinformatics) [45]. DESeq2 (v1.22.2) [46] was used for the differential expression analysis.

## Figures and Tables

**Figure 1 ijms-24-00148-f001:**
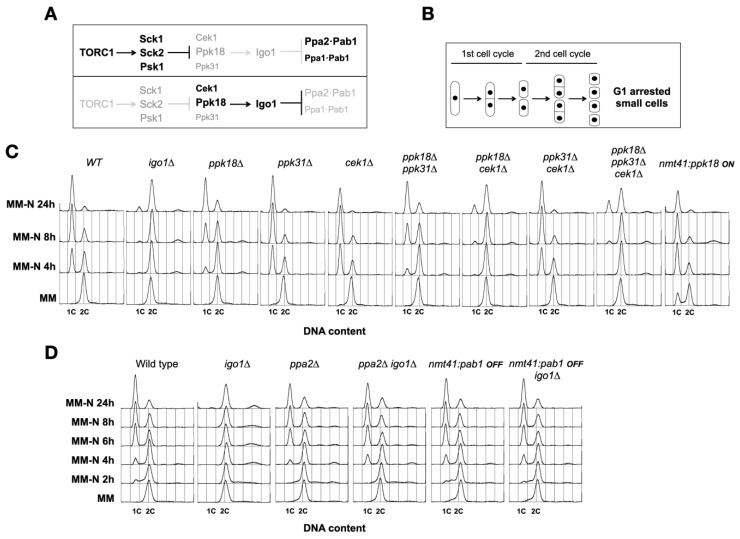
The Greatwall–Endosulfine switch is required for the last two cell divisions preceding G1 arrest in nitrogen-free medium (MM-N). (**A**) Schematic of the Greatwall–Endosulfine–PP2A/B55 pathway in fission yeast in nitrogen-rich media (above) and nitrogen-poor media (below). (**B**) Diagram of the two cell divisions that take place during nitrogen starvation. (**C**) FACS profile showing DNA content (1C and 2C) of wild type, *igo1*∆, *ppk18*∆, *ppk31*∆, *cek1*∆, *ppk18*∆ *ppk31*∆, *ppk18*∆ *cek1*∆, *ppk31*∆ *cek1*∆, *ppk18*∆ *ppk31*∆ *cek1*∆, and *nmt41:ppk18^+^* cells grown in minimal medium (MM) at 25 °C until mid-exponential phase and then transferred to MM-N. (**D**) FACS profile showing DNA content (1C and 2C) of wild type, *igo1*∆, *ppa2*∆, *ppa2*∆ *igo1*∆, *nmt41:pab1^+^, and nmt41:pab1^+^ igo1*∆ cells grown in MM containing thiamine (promoter OFF) at 25 °C until mid-exponential phase and then transferred to MM-N.

**Figure 2 ijms-24-00148-f002:**
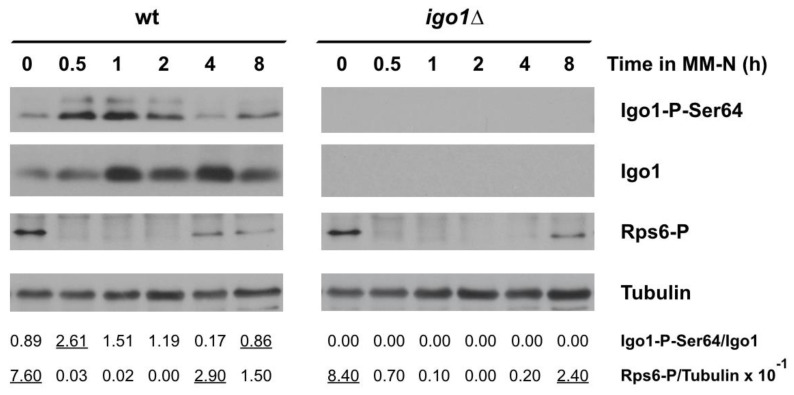
TORC1 and Greatwall activity oscillate out of phase during nitrogen starvation. Wild type and *igo1*∆ cells grown in minimal medium at 25 °C until mid-exponential phase were transferred to nitrogen-free minimal medium (MM-N) for 8 h at 25 °C. Samples were taken at 0, 0.5, 1, 2, 4, and 8 h in MM-N. Greatwall and TORC1 activities were determined using specific anti-phospho-Ser64-Igo1 and anti-Rps6-Ser235/236 antibodies. Igo1 protein levels were determined using anti-Igo1 antibodies. Tubulin was used as a loading control. The Igo1-P/Igo1 and Rps6-P/Tubulin ratios are indicated under the gels. Underlined numbers correspond to the peaks of activity.

**Figure 4 ijms-24-00148-f004:**
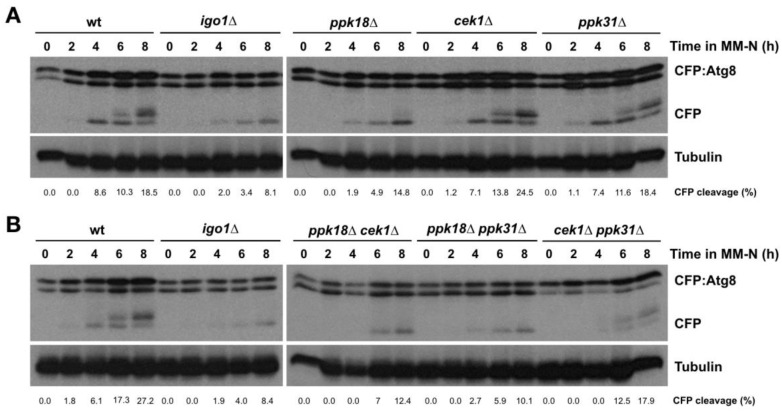
The Greatwall–Endosulfine switch promotes autophagy. (**A**) Wild type, *igo1*∆, *ppk18*∆, *cek1*∆, *and ppk31*∆ cells expressing CFP:Atg8 were grown in minimal medium at 25 °C until mid-exponential phase and then transferred to nitrogen-free minimal medium (MM-N) for 8 h at 25 °C. Samples were taken at 0, 2, 4, 6, and 8 h in MM-N. (**B**) Wild type, *igo1*∆, *ppk18*∆ *cek1*∆, *ppk18*∆ *ppk31*∆, and *cek1*∆ *ppk31*∆ cells expressing CFP:Atg8 were grown in minimal medium at 25 °C until mid-exponential phase and then transferred to nitrogen-free minimal medium (MM-N) for 8 h at 25 °C. Samples were taken at 0, 2, 4, 6, and 8 h in MM-N. CFP:Atg8 and free CFP levels were determined using anti-GFP antibodies (Clontech). Tubulin was used as a loading control. The percentage of CFP cleavage is indicated under the gels.

**Figure 5 ijms-24-00148-f005:**
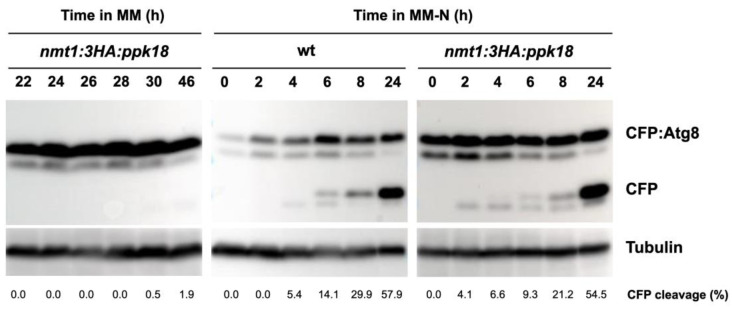
Greatwall overexpression is not sufficient to induce autophagy but slightly accelerates autophagic flux. Wild type and cells expressing *ppk18^+^* under the *nmt1* promoter *(nmt1:3xHA:ppk18^+^*) and CFP:Atg8 were grown in thiamine-containing minimal medium (MM+T, promoter OFF) at 25 °C until mid-exponential phase, and then transferred to thiamine-free minimal medium (MM-T, promoter ON) to induce *ppk18^+^* expression. After 22 h, part of the culture (panels middle and right) was transferred to nitrogen-free minimal medium (MM-N) for 24 h at 25 °C. Samples were taken at 0, 2, 4, 6, 8, and 24 h in MM-N. CFP:Atg8 and free CFP levels were determined using anti-GFP antibodies (Chromotek). Tubulin was used as a loading control. The percentage of CFP cleavage is indicated under the gels.

**Figure 6 ijms-24-00148-f006:**
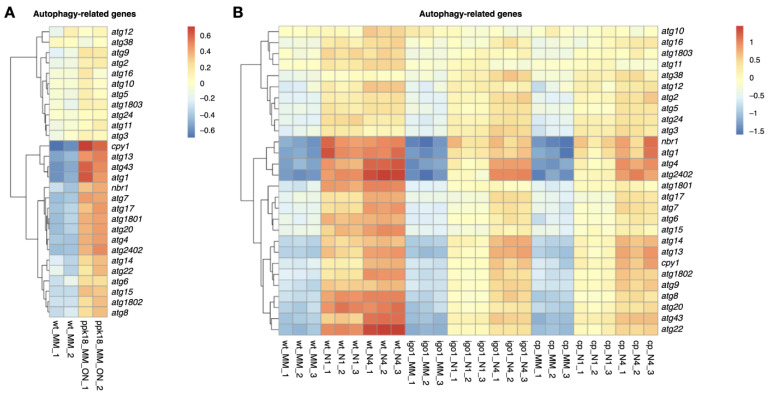
Heat maps of RNA-seq data showing autophagy-related genes induced by the Greatwall–Endosulfine switch. (**A**) Heat map showing autophagy-related genes upregulated by RNAseq after 20 h of Greatwall overexpression (*nmt1:ppk18^+^*) in minimal medium compared to the wild type. Two independent experimental replicates are shown. (**B**) Autophagy-related genes induced by nitrogen starvation in wild type (wt), *igo1*∆ (igo1), and *cek1*∆ *ppk18*∆ (cp) in MM, MM-N 1 h (N1), and MM-N 4 h (N4). Three independent experimental replicates are shown.

**Figure 7 ijms-24-00148-f007:**
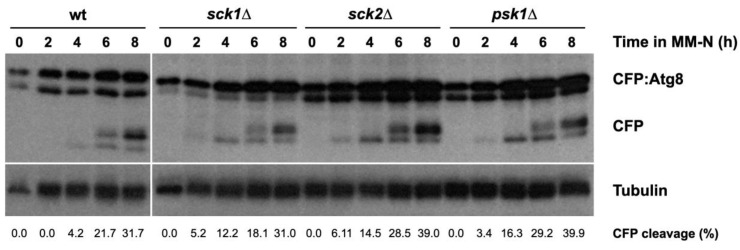
S6 kinases modulate autophagy. Wild type, *sck1*∆, *sck2*∆, and *psk1*∆ cells expressing CFP:Atg8 were grown in minimal medium at 25 °C until mid-exponential phase and then transferred to nitrogen-free minimal medium (MM-N) for 8 h at 25 °C. Samples were taken at 0, 2, 4, 6, and 8 h in MM-N. CFP:Atg8 and free CFP levels were determined using anti-GFP antibodies (Clontech). Tubulin was used as a loading control. The percentage of CFP cleavage is indicated under the gels.

**Figure 8 ijms-24-00148-f008:**
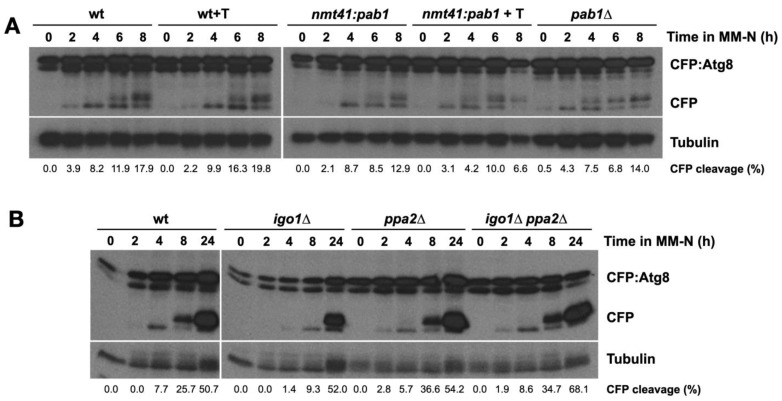
PP2A/B55 negatively regulates nitrogen starvation-induced autophagy. (**A**) Wild type, *nmt41:pab1^+^* cells, grown in minimal medium with (+T) or without thiamine at 25 °C, and *pab1*∆ cells grown until mid-exponential phase were transferred to nitrogen-free minimal medium (MM-N) for 8 h at 25 °C. Samples were collected at 0, 2, 4, 6, and 8 h in MM-N. CFP:Atg8 and free CFP levels were determined using anti-GFP antibodies (Clontech). Tubulin was used as a loading control. (**B**) Wild type, *igo1*∆, *ppa2*∆, and *igo1*∆ *ppa2*∆ cells grown in minimal medium at 25 °C until mid-exponential phase were transferred to nitrogen-free minimal medium (MM-N) for 24 h at 25 °C. Samples were collected at 0, 2, 4, 8, and 24 h in MM-N. CFP:Atg8 and free CFP levels were determined using anti-GFP antibodies (Clontech). Tubulin was used as a loading control. The percentage of CFP cleavage is indicated under the gels.

## Data Availability

The RNAseq data in this study have been deposited in the GEO database with the following accession number: GSE217398.

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
