# Peer review of "The Greatwall–Endosulfine Switch Accelerates Autophagic Flux during the Cell Divisions Leading to G1 Arrest and Entry into Quiescence in Fission Yeast"

_ijms, 2022, doi:10.3390/ijms24010148_

Round 1
Reviewer 1 Report
In this manuscript, Vázquez-Bolado et al., investigate the role of the Greatwall pathway in the autophagy flux induction in nitrogen-starved Schizosaccharomyces pombe cells. The Greatwall pathway regulates several mechanisms in different species. In budding and fission yeasts, its main role is to induce entry into quiescence. Entry into quiescence is characterized by the induction of a quiescence program, including autophagy induction. Here, this study presents lines of evidence of a role of the Greatwall-Endosulfine pathway in the timely initiation of the autophagy flux by controlling directly or indirectly the expression of some autophagy-regulated genes. While I find some of the data presented convincing, I believe the manuscript cannot be accepted in its present form and requires some revisions.
Major concerns:
Discrepancies between some figures brings some doubt:
- On the reproducibility of the experiments. Indeed the level of free CFP at 8 hours in igo1∆ is different between panels (compare figure 3 or figure 4A and figure 7C). For the WT, the level of the free form of CFP is strikingly different between panels (compare figure 3 and 7, and figure 4A, 4B and 5),
- On the validity of the conclusions. Indeed, the authors’ conclusion on the role the S6 kinases on the negative modulation of the autophagy flux relies on the accumulation of free CFP at 2hours (Figure 4B). However, for the WT, some free GFP can be detected already at 2 hours in figure 3 and in figure 7 but barely or not in figure 4A, 4B (panel used for the conclusion) and 5. A similar concern can be made on the effect of overexpressing ppk18+ from the nmt1 promoter.
The authors should either give some quantification of the free GFP for each condition or present more convincing and reproducible data.
As the conclusions of the authors rely on differences of free CFP observed at the early timepoints (2-4 hours) in MM-N medium (the differences seem attenuated at later timepoints, which could be confirmed (or infirmed) with quantifications), the authors should tone down their conclusions. For instance L186 or L251 add “slightly” to the sentences “an acceleration of the autophagic flux” or “autophagy flux increased”, respectively. In addition, rather than referring to an acceleration of the autophagy flux, a reference to a timely induction (or initiation) of autophagy would be more accurate (unless quantification are provided).
Minor:
Is there a reason why the authors did not analyze the processing of CFP:Atg8 in a igo1-S64A mutant? In ppk31∆ mutant alone or in combination with ppk18∆ and/or cek1∆?
L90: The authors indicate that the igo1∆ ppa2∆ or igo1∆ nmt41:pab1 double mutants displayed different phenotypes but they show only one (Figure 1D). They should either show additional phenotypes or refer to the ad hoc papers (at least the reference 16)
Figure 1A: It should be explained in the legend why Ppk18 and Ppa2-Pab1 are written in a larger size. What is the rationale to omit Sck1 and Psk1 while their role in timely induction of autophagy flux seems similar as that of Sck2.
L111-2, Figure 2A: The authors claim that the maximum level of phosphorylation of Igo1 at Serine 64 is observed at 1 hour after transferring cells to a nitrogen-free medium. As the total level of Igo1 increases as well between 30 minutes and 1 hour, the maximum level seems rather to be reached 30 minutes after the medium change. The authors should change their claim or provide more convincing data or give quantifications that support their claim. Moreover they should comment on the doublet detected with the anti phospho-Ser64-Igo1 antibodies.
L115, Figure 2A: The author claim there is a transient reactivation of TORC1 based on the detection of a phospho-form of Rps6 at 4 and 8 hours. As the total level of Rps6 is not shown, the claim is not conclusive. The author should at least show the total level of Rps6 and maybe give quantification or an additional time point (e.g. 10 hours) to show that the level of the phospho-form of Rps6 is really decreasing while that of phospho-Ser64-Igo1 is increasing.
L118-121: The author claim that the detection of a phospho-form of Rps6 at 8 hours in igo1∆ is the consequence of the reactivation of TORC1 due to the generation of recycled nitrogen. It is surprising that the low level of processed Atg8 from 4 to 8 hours in igo1∆ (figure 3) is sufficient to induce phosphorylation of Rps6 at the same level as the WT at 4 hours. The authors should comment this point.
L191-5: The overexpression of ppk18+ upregulates some autophagy-related genes in MM medium (Figure 6) but Atg8 is not processed (Figure 5). The authors should give an explanation.
Legend of Figures 5 and 6 and L186, L376 and L417. For the overexpression of ppk18+, the indicated time cells are grown in MM varies between 20 hours and 22 hours. The discrepancy should be fixed or explained.
L226-240: The authors show results obtained with pef1∆ and gad8∆ mutants. While the data is interesting for the field, it sounds a bit beyond the scope of the paper. It could be interesting to add a final model in which all the role of each player tested in the manuscript would be depicted to justify the rationale of testing them.
L292-3: In the reference 32, the authors used GFP:Atg8. The typo should be corrected.
L400: The authors use two types of anti-GFP antibodies. They should mention which one was used to detect the CFP and the GFP.
Supplementary figure 1A: The quality of the cytometry data is not as good as those presented in figure 1C and 1D. Could the authors give better data.
Supplementary figure 1B: The Nomarski images of WT cells grown 8 hours in MM-N show mainly small cells (likely G1 cells). However the facs data displayed in Supplementary figure 1A indicates that about 40 % of WT cells are still in G2 (Of note about 20% in figure 1C or 1D). It is rather confusing. The authors should indicate from which FACS data the Nomarski images correspond to.
Legend of supplementary figure 3: Fix the typos for the times at which the samples were taken upon the overexpression of ppk18+ in MM medium.
Reviewer 2 Report
Vazquez- Bolado et al report on their studies linking TORC1 regulated cell cycle progression with autophagy. Nitrogen withdrawal leads to inhibition of TORC1 signalling. This results in the activation of the Ppk18 and Cek1 kinases which in turn activate the endosulfine Igo1. Igo1 inhibits PP2A inducing accelerated progression into mitosis. The current study by xxx et al advances previous findings, that identified Ppk18, Cek1 and Igo1 as TORC1 targets required for cell cycle progression following nitrogen withdrawal.
Previous studies by Aono et al., (2019) identified the Ppk18 and Cek1 kinases as being required for cell cycle progression following TORC1 inhibition. Vazquez- Bolado et al., have demonstrated that the Ppk18 and Cek1 kinases play a secondary role in regulating the expression of autophagic genes. Thus, the induction of autophagy is suppressed in ppk18, cek1 and igo1 mutants. Conversely, overexpression of Ppk18 or PP2A inhibition results in enhanced induction of autophagy. Additionally, TORC2 appears to play a role in regulating TORC1 activity and autophagy induction.
The current study clearly demonstrates that Ppk18 and Cek1 regulate not only cell cycle progression but autophagy induction when cells are subjected to nitrogen withdrawal. My concern is that samples from wt and mutant cells were not in the same cell cycle phase, when autophagy induction was assayed. For instance, only 50% of igo1 mutants are in G1 following 8h of nitrogen withdrawal relative to wt cells. Thus Western blot assays comparing wt and igo1 mutants following nitrogen withdrawal examines two different cell populations in terms of cell cycle phase. The authors should address this issue in the discussion. It would be interesting to investigate the role of the Ppk18/ Cek1 signalling pathway in regulating the induction of autophagy in cells arrests in different cell cycle phases.
Round 2
Reviewer 1 Report
The authors have done a good job in addressing adequately in the current version my concerns. However, there are still a few typos to be fixed (L381-4), which I must confess had escaped my attention during the first reviewing of this article. Indeed, the dilution of the antibodies should be written as follows:
rabbit anti-P-Ser64-Igo1 (1:1,000), rabbit anti-phospho-(Ser/Thr) Akt Substrate for the phosphorylation of Rps6 (1:1,500) (Cell Signaling Technology), mouse anti-GFP (1:3,000) (Clontech, Figures 3, 4, 7 and 8), rat anti-GFP (1:5,000) (Chromotek, Figures 5, S2 and S3) and mouse anti-tubulin (1:1,0000)
Author Response
Answers to Reviewer #1 Comments (Round 2):
The authors have done a good job in addressing adequately in the current version my concerns. However, there are still a few typos to be fixed (L381-4), which I must confess had escaped my attention during the first reviewing of this article. Indeed, the dilution of the antibodies should be written as follows:
rabbit anti-P-Ser64-Igo1 (1:1,000), rabbit anti-phospho-(Ser/Thr) Akt Substrate for the phosphorylation of Rps6 (1:1,500) (Cell Signaling Technology), mouse anti-GFP (1:3,000) (Clontech, Figures 3, 4, 7 and 8), rat anti-GFP (1:5,000) (Chromotek, Figures 5, S2 and S3) and mouse anti-tubulin (1:1,0000)
Answer: Thank you very much for your comments and corrections. We have corrected the typos.